# Silver Nanoparticles with Liquid Crystalline Ligands Based on Lactic Acid Derivatives

**DOI:** 10.3390/nano9081066

**Published:** 2019-07-25

**Authors:** Tinkara Troha, Miroslav Kaspar, Vera Hamplova, Martin Cigl, Jaroslav Havlicek, Damian Pociecha, Vladimira Novotna

**Affiliations:** 1Institute of Physics of the Czech Academy of Sciences, 18221 Prague, Czech Republic; 2Faculty of Chemistry, University of Warsaw, ul. Zwirki i Wigury 101, 02-089 Warsaw, Poland

**Keywords:** silver nanoparticles, liquid crystals, self-assembly, plasmonic

## Abstract

We have prepared and studied silver nanoparticles functionalized with ligands based on lactic acid derivatives. Several types of hybrid systems that differed in the size of silver nanoparticles as well as the length of surface ligands were analyzed. Transmission electron microscopy (TEM) observation provided information about the size and shape of nanoparticles and proved good homogeneity of studied systems. By dynamic light scattering (DLS) measurements, we have measured the size distribution of nanoparticle systems. Plasmonic resonance was detected at around 450 nm. For two hybrid systems, the mesomorphic behaviour has been demonstrated by x-ray measurements. The observed thermotropic liquid crystalline phases reveal lamellar character. We have proposed a model based on self-assembly of intercalated liquid crystalline ligands.

## 1. Introduction

Noble metal nanoparticles (NPs) have been extensively studied in recent years due to their unique properties, which are different from those of the bulk material. Gold and silver NPs have attracted attention due to their optical properties related to plasmon resonance, an absorption peak that can be tuned by varying the NPs size, shape, spacing, and external dielectric environment [1,2,3,4,5,6]. This behavior makes them useful for numerous biological, optical and electronic applications [7,8]. Due to the strong enhancement of the electromagnetic field near the particles surface, they are also used as substrates for the surface-enhanced Raman spectroscopy [9,10,11]. Covering nanoparticles with organic surface ligands prevent aggregation and essentially governs interparticle spacing. Thus, the interparticle distance can be varied by the ligand length and precisely tuned at molecular level using different chemical moieties. Generally, such hybrid systems provide a model structure for photonic crystals as periodic structures composed of dielectric and metallic nanostructures, which affect the propagation of electromagnetic waves. The assembly of nanoparticles organized into a superlattice can provide a pathway to a large variety of materials [12]. A nanocrystal superlattice as an array of inorganic particles separated by layers of organic surface ligand enables novel biomedical and optoelectronic applications. Additionally, coating of the nanocrystals with organic ligands can lead to more elaborate nanoparticle assemblies with the possibility to keep control over various structures [13,14,15,16,17,18,19]. Within the last decade, gold and silver nanoparticles were synthesized and functionalized by thiols [20,21,22]. Still, there are questions concerning nanoparticle systems in which the organic part occupies a larger volume than that of the inorganic nanoparticles. The relationship between the nanoparticle size and interactions between ligand molecules is not yet fully explained. To keep control over the structure of superlattices consisting of functionalized nanoparticles remains a challenge.

In the present contribution, we have prepared and studied silver nanoparticles functionalized with lactic acid derivatives. Most of the previously described procedures are based on two-step shielding, with the first step applying only short alkyl-thiols [13,14,15,16,17,18,20,23,24]. In the second step, a part of short thiols on the nanoparticle surface is exchanged by longer mesogenic molecules. In contrast to these procedures, we have applied mesogenic ligands directly in the process of nanoparticle synthesis. To the best of our knowledge, such a direct method to prepare functionalized silver nanoparticles has not yet been accomplished. Our aim is to enrich the number of mesogenic molecules during the process of functionalization. Therefore, we have applied mesogenic chiral rod-like thiols directly in preparation of hybrid systems. We have tuned the molecular structure of ligands and varied the size of silver nanoparticles. Generally, lactic acid derivatives were found to exhibit good thermal stability and optical purity for preparation of thermotropic liquid crystals. Previously, we have studied lactic acid derivatives with a similar molecular structure exhibiting mesomorphic properties, and, for selected types of mesogenic molecules, the smectic phases were present in broad temperature ranges [13,14,15,16,18,20,24]. Moreover, the chirality of organic ligands can support the creation of chiral superstructures.

## 2. Synthesis of Ligand and the Silver Nanoparticle Preparation

First of all, we have synthesized three types of thiols as ligands, with different molecular cores, to study the relationship between the size of Ag nanoparticles and the organic ligand length with respect to properties of new hybrid systems. Chemical formulae of all studied ligands are in Figure 1. Synthesis scheme and details of characterization are in the electronic supporting info file (Appendix A). For the first type of ligands, we applied the shortest mesogenic thiol, which was designated Thiol A. The ligand molecules consisted of two benzene rings in the molecular core, the non-chiral chain is terminated with a thiol and the chiral chain contains the lactate unit (Figure 1a). The molecular length was established to be ~2.5 nm. The scheme of Thiol A synthesis is in Appendix A and corresponding NMR (nuclear magnetic resonance) spectra in Appendix A.

The second ligand Thiol B was longer, the molecular core consists from three phenyl rings (the length is ≈ 3.5 nm), and its chemical formula is shown in Figure 1b. The scheme of synthesis is in Appendix A. The third ligand Thiol C has the longest molecule, its molecular core is composed of two biphenyls linked via an ester linkage, and one biphenyl is laterally substituted by a chlorine atom. By molecular simulations, we can establish the molecular length for Thiol C ≈ 4.5 nm. The study of self-assembling properties of thiols is rather tricky because they are very reactive, which does not allow us to determine the character of mesophases in details. We have established that Thiol B and Thiol C revealed tendency to create mesophases. Relevant differential scanning calorimetry (DSC) thermograms for Thiol B are presented in Appendix A.

Silver nanoparticles were synthesized by reduction of silver nitrate solutions. Namely, a modified Creighton method [25], and the two green synthetic methods using either tannin [26] or sucrose [27] as reducing agents were also tested. After modifications and optimization of the procedures, all methods gave almost the same range of particle size. The NP1 system capped with Thiol A ligand was synthesized using the tannin method. The second type of nanoparticles capped with Thiol B (system NP2) was prepared using the Creighton borohydride method. Finally, the systems NP3 and NP4 were synthesized using the sucrose method in the presence of Thiol C, differing only in the final size of the nanoparticles. Details of experimental procedures are presented in Appendix A.

## 3. Experimental Techniques

Transmission electron microscope TEM Libra 120 from Zeiss (Oberkochen, Germany) is used with the highly flexible Koehler illumination system, together with the in-column OMEGA filter, providing an excellent image contrast. Thermogravimetric differential thermal analysis (TG-DTA) measurements were performed with thermal analyzer Setaram Setsys Evolution (Setaram, France), operating range of 25–1600 °C.

Dynamic light scattering (DLS) was accomplished using Nd-YAG laser as a light source with the wavelength of 532 nm. Intensity correlation function *g*^(2)^(τ) = <*I(t) I(t+τ)*>/<*I(t)*><*I(t+τ)*> was measured using ALV5000 autocorrelator (ALV Laser Vertriebsgesellschaft m.b.H., Langen, Germany) enabling measurements over the time range of 10^−8^ to 10^3^ s.

Differential scanning calorimetry (DSC) has been performed using Perkin-Elmer 7 calorimeter (Perkin Elmer, Inc., Waltham, MA, USA). The materials, about 2–5 mg, were hermetically sealed in aluminium pans and inserted into the calorimeter. A nitrogen atmosphere was utilized during measurements. Temperatures and enthalpy changes were calibrated on extrapolated onset temperatures and enthalpy changes for melting points of water, indium, and zinc. Calorimetric studies were done on cooling/heating runs at a rate of 10 K/min.

X-ray studies have been performed using the Bruker D8 Discover system (CuKα radiation, Bruker, Billerica, MA, USA) equipped with an Anton Paar DCS-350 heating stage (temperature stability 0.1 K, Anton Paar GmbH, Graz, Austria), working in the reflection mode. Samples were prepared as one-surface-free droplets on a heated surface. The smectic layer thickness, *d*, was determined using the Bragg’s law nλ = 2dsin*θ*.

UV-Vis spectroscopy was performed with a Perkin Elmer Lambda 1050 (Perkin Elmer, Inc., Waltham, MA, USA) high-performance spectrophotometer in the wavelength range 250–1200 nm.

## 4. Results and discussion

### 4.1. Nanoparticle Observation

Silver nanoparticles (AgNPs) of spherical shape were prepared according to the above-described procedures (see Appendix A for details), and we were able to separate them with respect to their size. The preparation of organic ligands is described in detail in Appendix A. Liquid crystalline (LC)-ligands (Thiol A, Thiol B, or Thiol C) were chemically attached to the nanoparticles’ surface. The resulting structures create different nanoparticle systems, which were labelled as NP1, NP2, NP3, and NP4. The size of the nanoparticles was first detected by x-ray measurements, details about size distribution were observed under the transmission electron microscope (Figure 2 and Appendix A), and for selected systems, the distribution of sizes was established from DLS measurements. In Table 1, the parameters of studied systems are presented. We have prepared two systems with Thiol C. The nanoparticle systems NP3 and NP4 were prepared by the same procedure. The only modification was that, for NP4, the mixture of starting solutions (see Appendix A) was poured into distilled water with ice to perform the reaction at a lower temperature. This procedure probably influenced the growth of nanoparticles, and we succeeded in reaching the smaller size. In any case, we have studied all systems to be able to compare and evaluate the relationship between the nanoparticle size and length of ligand.

### 4.2. Dynamic Light Scattering (DLS) Measurements

Dynamic light scattering experiment was performed in order to determine the size distribution of nanoparticles. In this experiment, light scatters on molecules that, due to their Brownian motion, cause the fluctuations of the scattered light intensity. The time scale of these fluctuations is correlated to the size of the particles. In our experiment, the measured signal was a consequence of the single nanoparticles and also of the flocculates or aggregates that formed in the solution. In this case, the electric field autocorrelation function *g*^(1)^(τ) = <*E(t) E*(t+τ)*>/<*E(t)*><*E*(t+τ)*> is proportional to the following term [28]:(1)$$g^{\left(1\right)}\left(q,\tau \right)\propto \int P\left(R\right)R^{6}{\left|b\left(q,R\right)\right|}^{2}e^{-D\left(R\right)q^{2}\tau }dR$$
where *P*(*R*) is the number distribution of the particle radii, *b* is the form factor of the single particle, *D* is diffusion constant, and *q* the magnitude of the scattering wave vector. From the equation, we see that the signal coming from the particle is proportional to the sixth power of its radius, and therefore the smaller particles are often screened by the larger ones. In other words, the detection of the small particles in the presence of the large ones is hindered. In order to determine *P*(*R*) from the time dependence of the autocorrelation function *g*^(1)^(τ), we used the CONTIN algorithm [29].

Samples for measurements were prepared by diluting the studied composites in toluene to the final concentration 50 µg/mL. Unfortunately, we were not able to dissolve NP1 sample in toluene, probably agglomerates were present. Therefore, we have analyzed three samples NP2, NP3, and NP4. The result of the measurement was intensity autocorrelation function *g*^(2)^(τ) = <*I(t) I(t+τ)*>/<*I(t)*><*I(t+τ)*>, from which we calculated the electric field autocorrelation function *g*^(1)^(τ) = <*E(t) E*(t+τ)*>/<*E(t)*><*E*(t+τ)*> by taking into account the static light scattering contribution [28]. The results are presented in Figure 3a. Using CONTIN algorithm, the decay times were calculated and are shown on Figure 3b. Sample NP3 shows two relaxation times at t_1_ = 7 ms and t_2_ = 0.04 ms. Similarly, two relaxation times of NP4 are t_1_ = 4 ms and t_2_ = 0.03 ms. Sample NP2 shows three relaxation times, t_1_ = 4 ms, t_2_ = 0.13 ms, and t_3_ = 0.01 ms. The shortest relaxation time corresponds to the Brownian motion of the single nanoparticle, and the larger ones correspond to the motion of flocculates or aggregates. Figure 3c shows the number distribution of the hydrodynamic radii, *R*, for the single nanoparticles. Their average radii are 4 nm for NP2, 10 nm for NP3, and 6 nm for NP4.

### 4.3. Plasmonic Resonance

Samples were diluted in toluene solution, and mechanical sonification of the colloidal suspension was performed to support the dissolving process. During this process, the solution was heated and kept at 60 °C. A droplet of the solution was deposited on a sapphire plate and dried in air. The UV-VIS spectra were measured using the Perkin Elmer spectrometer in the wavelength range between 300 and 800 nm. Figure 4 shows UV-visible absorption spectrum of the sample N2. An absorption peak at around 450 nm is associated to surface plasmonic resonance, which is caused by the collective oscillations of the electrons in the nanoparticles. The other two samples did not show clear peaks for the plasmonic resonance because the absorption spectra were more smeared out. Nevertheless, a broad maximum is observed within the range 420–550 nm. In addition, for NP2, we observed a blue shift of the plasmonic resonance peak for 8 nm after the heating from the room temperature to the temperature 100 °C, at which the sample is in the liquid crystalline phase. According to theoretical explanations [30], the plasmon coupling occurs when nanoparticles approach each other for a distance less than their diameter. Then, the plasmonic resonances of individual particles start to couple, which results in red shift of the plasmonic peak wavelength. In our case, the blue shift was observed, indicating that the interparticle distance in smectic phase is increased in comparison to the crystalline state. 

### 4.4. Mesomorphic Properties

We have analysed all our samples with respect to the temperature behaviour. Thermogravimetric analysis (TGA) was performed in order to investigate thermal stability of the samples (Figure 5). It was observed that the thermal stability of NP2 sample is not optimal because the sample degradation process started already above 150 °C. On the contrary, the other two samples, NP3 and NP4, exhibit better thermal stability and reveal higher degradation temperatures—for NP3, it is at around 270 °C, and for NP4 at around 300 °C.

We have studied our samples by the DSC method and under polarizing microscope in a wide temperature range, taking into account information from TGA measurements. We have found that two samples, NP2 and NP4, reveal a melting point on heating at about 50–70 °C, and for higher temperatures, a self-assembling behaviour was observed. Figure 6 shows the second heating (red colour) and cooling run (blue colour) detected during the DSC measurements for NP4. The crystallization is not fully accomplished on the first heating and additional crystallization peak appeared on subsequent heating in the opposite sense (being exothermic instead of endothermic). The melting is split into two peaks, which are probably due to presence of several crystalline states. The results of DSC measurements for the sample NP2 are shown in Appendix A. Corresponding enthalpies and onset of phase transition temperatures taken from DSC data are summarized in Appendix A.

We have observed liquid crystalline (LC) properties under a polarizing microscope. On heating from the crystalline state, we can see composite systems melt at a temperature of about 60 °C for NP2 and NP4. Between two glass plates, we have observed texture that can be found for smectic A phase in homeotropic anchoring of molecules (Figure 7a), which is black under crossed polarizers due to predominant orientation of the optical axis. Unfortunately, agglomerates are also present in the LC phase. They are visible as black spots in the microscope when we observe in non-polarized light (Figure 7b). Our observations confirm experimental experience [31] that homeotropic anchoring prevails for hybrid systems and mixtures of NPs with LC compounds. When the shear was applied, the alignment changed, and birefringence colour appeared. The modified texture after the application of the shear is shown in Figure 7c for NP4. We confirmed liquid crystalline properties of NP2 and NP4 systems in broad temperature intervals in agreement with DSC and x-ray techniques (see later).

We have dissolved NP4 in a toluene and prepared a freely-evaporated film on a glass substrate. The crystallinity at the room temperatures was proved by atomic force microscopy (AFM) measurements. AFM picture at the room temperature is shown in Appendix A. On heating above the melting temperature, the film melted to LC state which is too soft to be analysed by AFM.

X-ray scattering is a powerful tool for material characterization. The elastic scattering is determined by two parameters: the form factor characterizing shape and size of nanoparticles, and the structure factor, which reflects the spatial arrangement of objects. The form factor dominates for larger nanoparticles, and the size of nanoparticles can be evaluated. For the NP3 system, the size of the nanoparticles was established as 18 nm. Unfortunately, for small nanoparticles, the form-factor analysis is not precise enough.

For systems NP2 and NP4, we have measured x-ray scattering on heating and cooling and checked the presence of a liquid crystalline phase. On heating above the melting temperature, a smectic-like structure was detected in both systems. For NP2, the x-ray intensity versus scattering angle is presented in Figure 8, and commensurate periodicity is clearly demonstrated. We have analysed x-ray scattering data and found that the periodicity corresponds to about 8 nm. Unfortunately, the thermal stability of NP2 sample under x-ray illumination is limited and decomposition occurred on heating above 150 °C. The x-ray intensity profile versus scattering angle and temperature was measured on the first heating, and results are shown in Appendix A. On subsequent cooling, NP2 sample is decomposed, so x-ray measurements did not provide reasonable data.

On the other hand, for sample NP4, the x-ray measurements are reproducible on heating/cooling runs and clearly show the presence of a smectic structure with a liquid-like organization within layers even on a subsequent cooling from the isotropic phase. On heating we have observed the melting process from the crystalline state at about 58 °C, then at wide-scattering angle the sharp signals corresponding a crystalline phase disappeared. At higher temperatures, a smectic mesophase was detected within a broad temperature interval. A series of commensurate signals were observed in small-angle range indicating a lamellar structure. In a wide-angle region, only a diffused signal was present, which evidences liquid-like correlation between ligands within the smectic layers. On heating above 180 °C, the isotropic phase was reached, and the sharp signal in the small-angle region (from the lamellar structure) faded away. In Appendix A, we present a profile of the intensity versus the scattering angle for selected temperature in the smectic phase and in the isotropic phase for comparison to demonstrate this behaviour. In Figure 9, we show the x-ray pattern measured for NP4 on cooling from the isotropic phase through the smectic to crystalline phase. Below the temperature T = 55 °C, the signal at wide angles starts to split into sharp peaks, which indicates the crystalline character.

## 5. Conclusions

We have developed new procedures for preparation of silver nanoparticles. We have synthesized several types of chiral rod-like thiols, varying their length by prolongation of the molecular core. The thiols were applied for functionalization of silver nanoparticles and four nanoparticle systems were prepared, differing in nanoparticle size and the length of thiol molecules used as capping agents. We have characterized the prepared systems with TEM, TGA, DLS, and spectroscopic techniques.

In the first type of hybrid system—NP1—with the shortest thiol as a ligand, we were not successful, and agglomeration of nanoparticles was observed. For sample NP3 with large silver nanoparticles (20 nm), the ligand was too short to support self-assembling properties. On the contrary, for samples NP2 and NP4 with the small silver nanoparticles (4–6 nm) functionalized by long ligands (3–5 nm), we observed lamellar liquid crystalline phases. As the periodicity of such a structure is about 8 nm, we propose the model of layers with intercalated organic ligands as it is schematically depicted in Figure 10.

## Figures and Tables

**Figure 1 nanomaterials-09-01066-f001:**
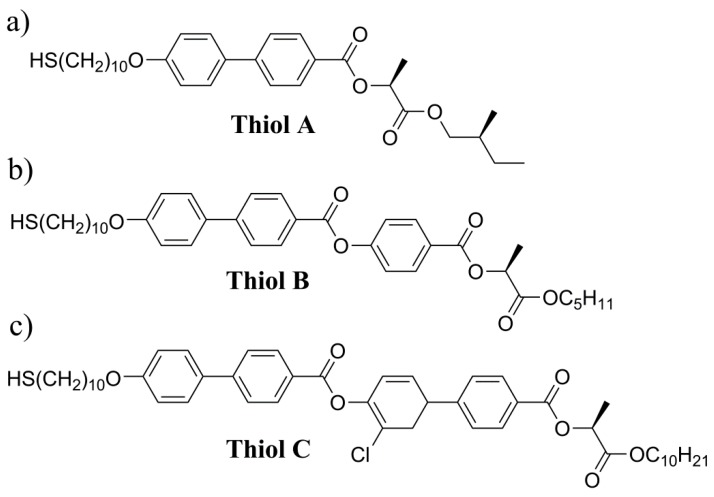
Chemical formulas of three different thiols (**a**) Thiol A (**b**) Thiol B and (**c**) Thiol C utilized as ligands for silver nanoparticles.

**Figure 2 nanomaterials-09-01066-f002:**
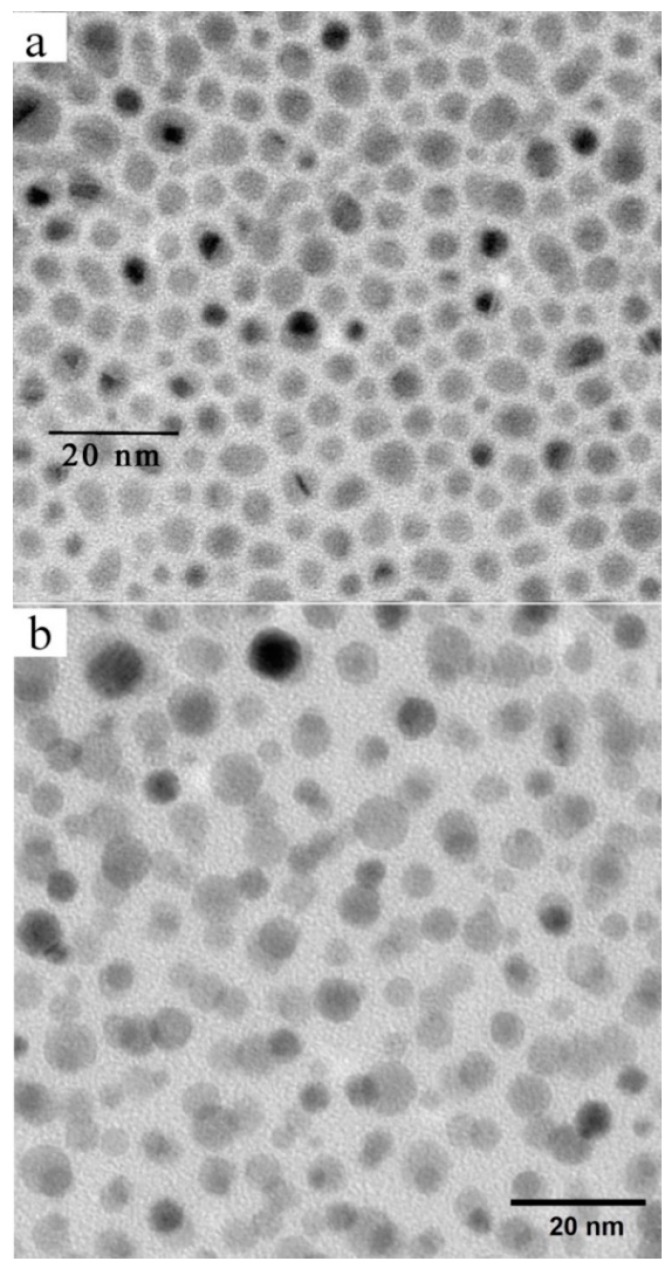
Transmission electron microscopy (TEM) figures for (**a**) NP1 and (**b**) NP3.

**Figure 3 nanomaterials-09-01066-f003:**
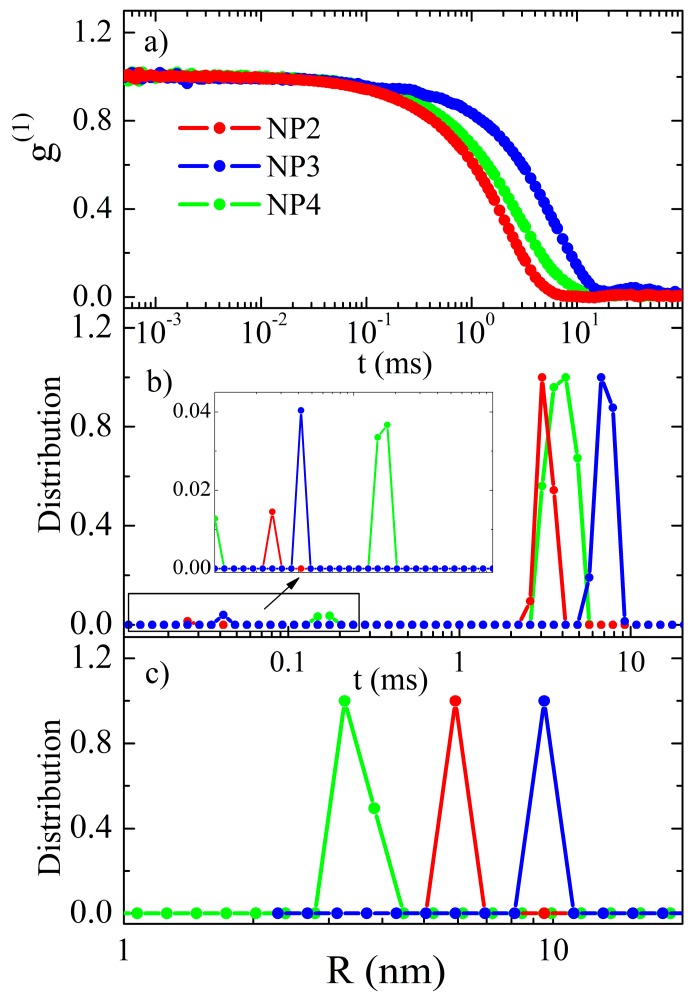
(**a**) Electric field autocorrelation function *g*^(1)^ in the time scale, (**b**) distribution of decay times, and (**c**) distribution of the hydrodynamic radii.

**Figure 4 nanomaterials-09-01066-f004:**
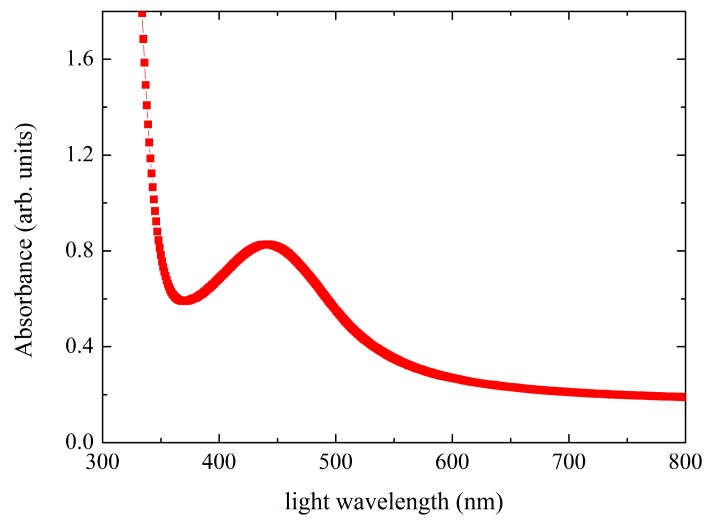
UV-VIS spectrum of N2 sample prepared as film on sapphire platelet.

**Figure 5 nanomaterials-09-01066-f005:**
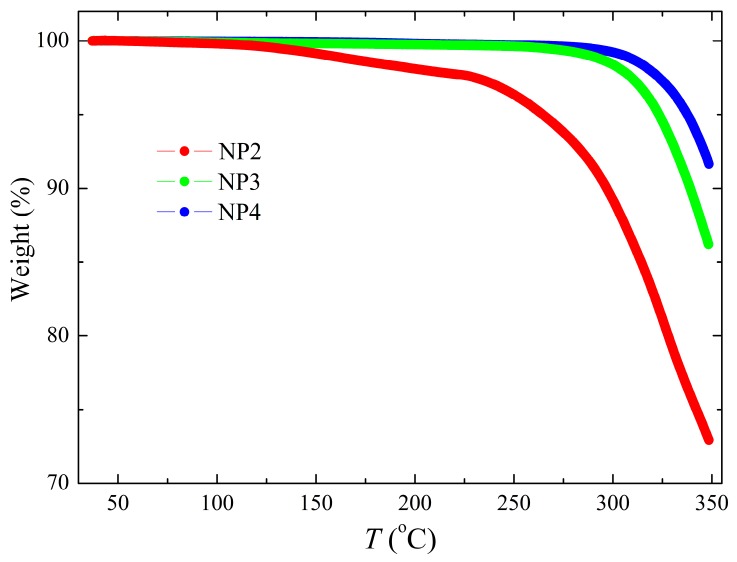
Thermogravimetric analysis (TGA) showing weight in % as a function of temperature for the samples NP2, NP3, and NP4.

**Figure 6 nanomaterials-09-01066-f006:**
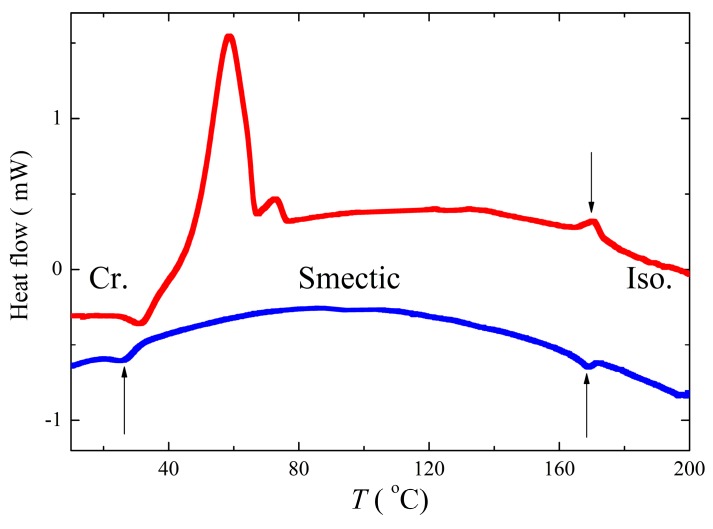
Differential scanning calorimetry (DSC) plots for NP4 are presented: the second heating (red colour) and cooling (blue) runs, the phase transitions are marked by arrows, and the phase type is depicted in corresponding temperature ranges.

**Figure 7 nanomaterials-09-01066-f007:**
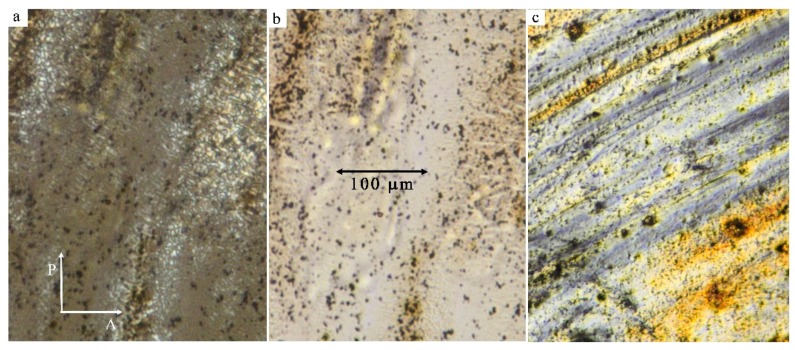
Texture of NP4 between two glass plates: (**a**) a homeotropic texture under crossed polarizers, (**b**) without polarizers, and (**c**) the texture under crossed polarizers after application of the shear.

**Figure 8 nanomaterials-09-01066-f008:**
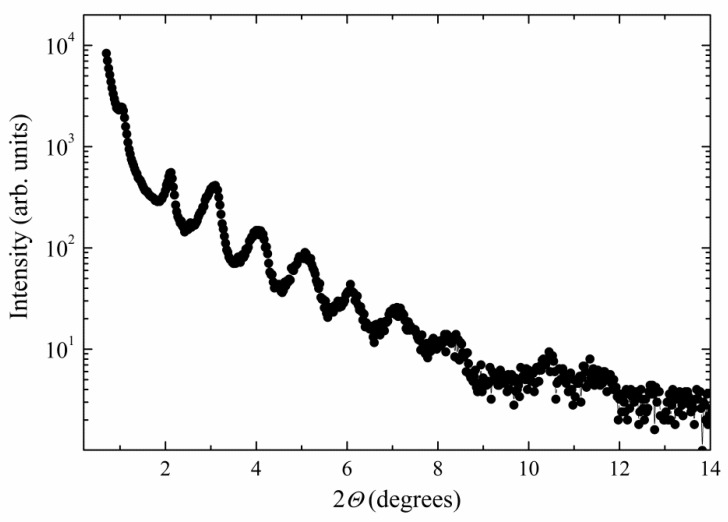
X-ray intensity versus scattering angle, *θ*, for NP2 sample.

**Figure 9 nanomaterials-09-01066-f009:**
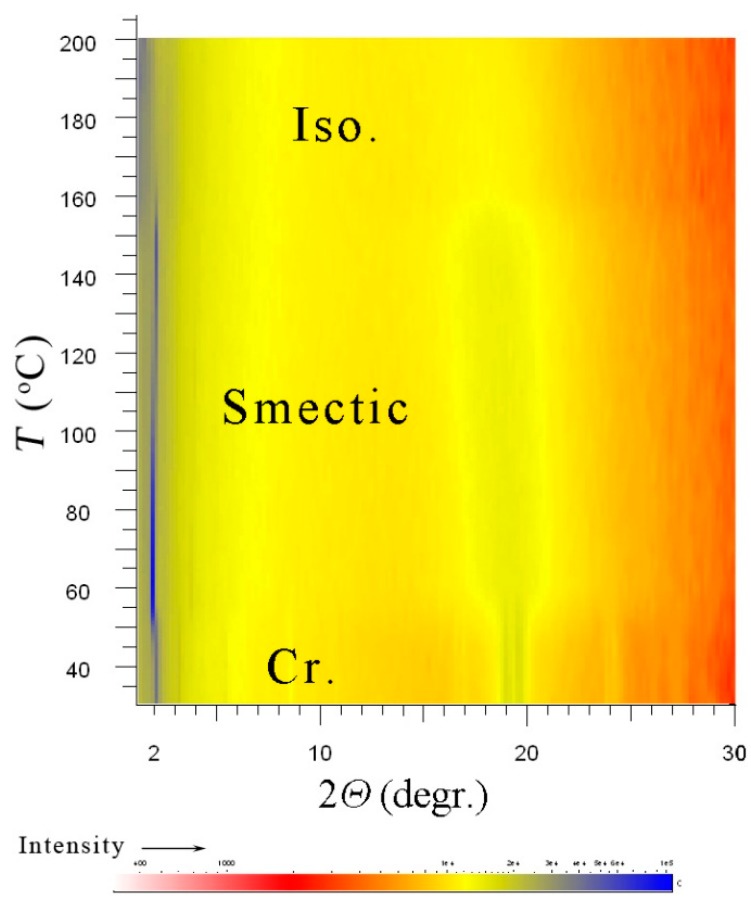
Temperature dependences of the scattered x-ray intensity (corresponding colours are below the graph) for compound NP4 measured on cooling. The figure shows a distinct smectic-like mesophase between the isotropic liquid and the crystalline phases.

**Figure 10 nanomaterials-09-01066-f010:**
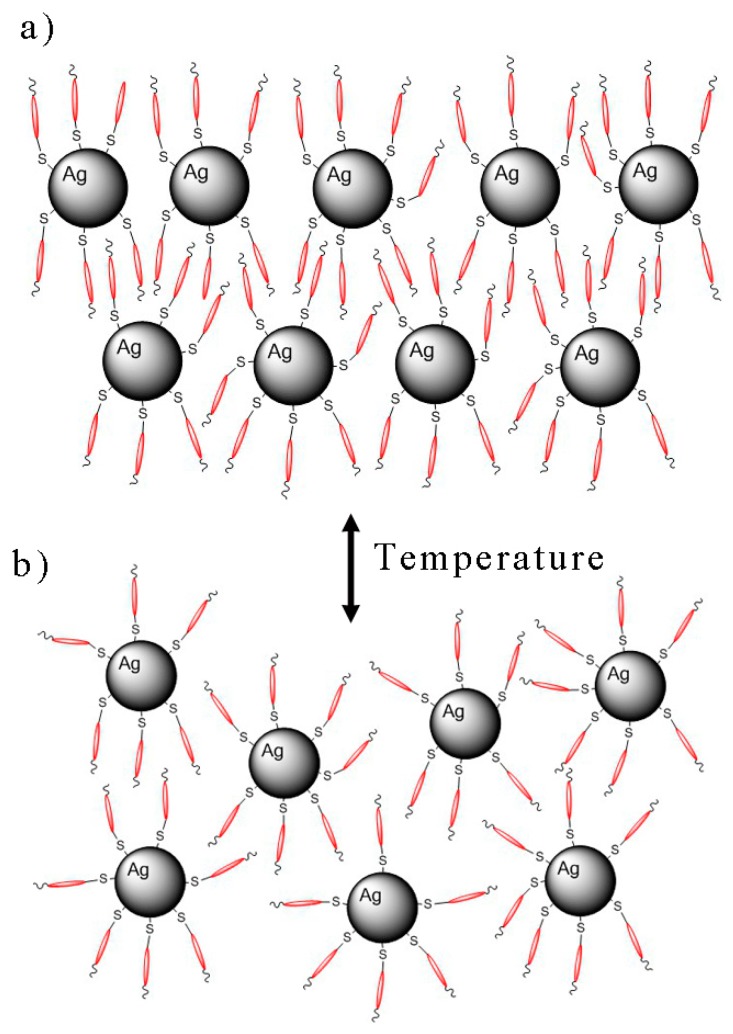
Schematic organization of nanoparticle system (**a**) packed in layered mesophase and (**b**) in the isotropic phase.

**Table 1 nanomaterials-09-01066-t001:** Designation of studied hybrid system with respect to attached ligand and diameter of silver nanoparticles (AgNPs), established by experimental techniques described later.

	LC-Ligand	AgNPs‘ Diameter
NP1	Thiol A	4–6 nm
NP2	Thiol B	3–5 nm
NP3	Thiol C	~20 nm
NP4	Thiol C	4–6 nm

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
