# Peer review of "Silver Nanoparticles with Liquid Crystalline Ligands Based on Lactic Acid Derivatives"

_nanomaterials, 2019, doi:10.3390/nano9081066_

Reviewer 1 Report

Authors reported on synthesis and characterization of silver nanoparticles functionalized with mesogenic ligands. They introduced mesogenic ligands directly in the process of nanoparticle synthesis. Nanoparticle size distribution was confirmed by DLS and TEM analysis. Liquid crystalline self-organization properties of hybrid nanoparticles were observed upon heating.

Herein, “Discussion” section is separated from “Results” section.  In this journal, authors should discuss the results and how they can be interpreted in perspective of previous studies and of the working hypotheses. The findings and their implications should be discussed in the broadest context possible and limitations of the work highlighted. However, the discussion section of this paper is seriously lacking in its criteria and rather close to the conclusions section. I recommend the authors consider reorganizing sections for “Results and Discussion” and “Conclusions”.  

Written below is comments and suggestions for further improvement.

Figure numbers are not designated in the main text. In addition, Page 6 line 169, (Fig.4) should be corrected to (Fig.5).

In Figure6, in the second cooling (red line), DSC curve shows two peaks between 40 and 80 deg. Authors discussed that hybrid silver nanoparticles show only lamellar smectic-like phase, but it seems to contain more than one phase. I wonder the small transition is from chiral reorganization?

In table 1, Why NP3 shows around 5 times larger nanoparticle size than that of NP4 though same ligand (Thiol C) is utilized? And only the nanoparticle size is related to the thermal stability difference?

Page 8 line 196, it does not make sense that material decomposes upon subsequent cooling. Please verify the sentence.

From Figure 8, it is hard to tell the difference between smectic and crystalline phase. In particular, intensity scale with colors and index is too faint to distinguish. Presenting at least one original XRD pattern showing smectic phase is suggested.

 Author Response

We thank the first referee for his/her careful reading of our manuscript and for valuable remarks and suggestions. We made changes by red color and tried to do our best to improve the text.

In detail what we did:

We have tried to enlarge the discussion attached to results presentation. We renamed Results to Results and discussion.

Figure numbers are now properly designated in the text. Page 6 line 169, (Fig.4) was corrected to (Fig.5).

We can explain double-peak for melting as melting from non-homogeneously crystallized state. On cooling in a dynamic regime the crystallization was probably not fully accomplished or there are two crystalline phases. The crystallization temperature generally depends on the rate, thermal history etc. for organic compounds. It is only information to prove we are not in glassy state, how was proposed by the second referee.

 We have tried to explain why NP3 shows larger nanoparticle size. We studied both systems, NP3 and NP4 to prove that for liquid crystalline packing the ligands should be longer or at least comparable in the length with the size on nanoparticles to support self-assembling processes.

Page 8 line 196, we have reformulated the sentences.

Reviewer 2 Report

The Authors present an experimental investigation concerning silver nanoparticles coated with mesogenic units. They report the synthesis and characterization of the nanoparticles and the characterization of their mesomorphic behaviour.

The work is interesting, I have no major comments on the first part, the synthesis and characterization of nanoparticles, which is well conducted.

However, the characterization of the smectic phases is based mostly on the X-ray pattern. Although x-ray are very important, the real signature for a liquid crystal phase identification is in the polarized light textures observed at the microscope.

I recommend to run some investigations of the texture to confirm that the  phase assignment is correct, that is a fluid smectic  phase rather than a glass or some other kind of disordered crystal. Moreover, the Authors showed a DSC but they did not report a Table with the values of the transition temperatures and, more important, the enthalpy changes at the transition. These data can be obtained from the DSC trace and they are also  are very important in order to assess the type of phase.

In short, although the Authors successfully report the synthesis and characterization of silver nanoparticles with mesogenic molecules they do not provide sufficient evidence of the liquid crystalline behaviour of their materials.

Author Response

We thank the second referee for his/her careful reading of our manuscript and for valuable remarks and suggestions. We made changes by red color and tried to do our best to improve the text.

In detail what we did:

We have added AFM picture to ESI to prove the crystalline character at the room temperature. From DSC the crystallization can be demonstrated as well.

We have added Table S1 to ESI and collected the onset phase transition temperatures and corresponding enthalpies into the table.

We have modified the text to explain the x-ray data in details. We added Figure S7 to ESI to present a profile of the scattering angle at one distinct temperature in the smectic phase and in the isotropic phase to demonstrate the difference and to prove that the signal from layers is strong enough to confirm the smectic phase presence.

Round  2

Reviewer 1 Report

This manuscript has been largely improved enough to be accepted in present form.

Reviewer 2 Report

The Authors have revised the manuscript according to the referee's suggestions. It can be now published in its present form.